# IGF1R Is a Potential New Therapeutic Target for HGNET-BCOR Brain Tumor Patients

**DOI:** 10.3390/ijms20123027

**Published:** 2019-06-21

**Authors:** Nadine Vewinger, Sabrina Huprich, Larissa Seidmann, Alexandra Russo, Francesca Alt, Hannah Bender, Clemens Sommer, David Samuel, Nadine Lehmann, Nora Backes, Lea Roth, Patrick N. Harter, Katharina Filipski, Jörg Faber, Claudia Paret

**Affiliations:** 1Section of Pediatric Oncology, Children’s Hospital, University Medical Center of the Johannes Gutenberg University Mainz, 55131 Mainz, Germany; Nadine.Vewinger@unimedizin-mainz.de (N.V.); sabrina.huprich@gmx.de (S.H.); Alexandra.Russo@unimedizin-mainz.de (A.R.); Francesca.Alt@unimedizin-mainz.de (F.A.); Bender.hannah@gmail.com (H.B.); Nadine.Lehmann@unimedizin-mainz.de (N.L.); Nora.Backes@unimedizin-mainz.de (N.B.); Lea.Roth@unimedizin-mainz.de (L.R.); Joerg.Faber@unimedizin-mainz.de (J.F.); 2Institute of Pathology, University Medical Center of the Johannes Gutenberg University Mainz, 55131 Mainz, Germany; larissa.seidmann@unimedizin-mainz.de; 3University Cancer Center of the Johannes Gutenberg University Mainz, 55131 Mainz, Germany; 4Institute of Neuropathology, University Medical Center of the Johannes Gutenberg University Mainz, 55131 Mainz, Germany; clemens.sommer@unimedizin-mainz.de; 5Department of Oncology, Valley Children’s Hospital, Madera, CA 93636, USA; DSamuel@valleychildrens.org; 6Neurological Institute (Edinger-Institute), Goethe-University Medical School, 60528 Frankfurt am Main, Germany and German Cancer Consortium (DKTK), Partner Site Frankfurt/Mainz, 69120 Heidelberg, Germany, German Cancer Research Center (DKFZ), 69120 Heidelberg, Germany; patrick.harter@kgu.de (P.N.H.); Katharina.Filipski@kgu.de (K.F.); 7Frankfurt Cancer Institute (FCI), 60596 Frankfurt am Main, Germany; 8German Cancer Consortium (DKTK), site Mainz, Germany, German Cancer Research Center (DKFZ), 69120 Heidelberg, Germany

**Keywords:** HGNET-BCOR, IGF1R, IGF2, ceritinib, vinblastine

## Abstract

(1) **Background:** The high-grade neuroepithelial tumor of the central nervous system with BCOR alteration (HGNET-BCOR) is a highly malignant tumor. Preclinical models and molecular targets are urgently required for this cancer. Previous data suggest a potential role of insulin-like growth factor (IGF) signaling in HGNET-BCOR. (2) **Methods:** The primary HGNET-BCOR cells PhKh1 were characterized by western blot, copy number variation, and methylation analysis and by electron microscopy. The expression of *IGF2* and *IGF1R* was assessed by qRT-PCR. The effect of chemotherapeutics and IGF1R inhibitors on PhKh1 proliferation was tested. The phosphorylation of IGF1R and downstream molecules was assessed by western blot. (3) **Results:** Phkh1 cells showed a DNA methylation profile compatible with the DNA methylation class “HGNET-BCOR” and morphologic features of cellular cannibalism. *IGF2* and *IGF1R* were highly expressed by three HGNET-BCOR tumor samples and PhKh1 cells. PhKh1 cells were particularly sensitive to vincristine, vinblastine, actinomycin D (IC_50_ < 10 nM for all drugs), and ceritinib (IC_50_ = 310 nM). Ceritinib was able to abrogate the proliferation of PhKh1 cells and blocked the phosphorylation of IGF1R and AKT. (4) **Conclusion:** IGF1R is as an attractive target for the development of new therapy protocols for HGNET-BCOR patients, which may include ceritinib and vinblastine.

## 1. Introduction

The high-grade neuroepithelial tumor with BCOR alteration (HGNET-BCOR) is a rare brain tumor first described as a distinct entity in 2016 [1]. HGNET-BCOR represents 3% of tumors with a prior diagnosis of “Primitive neuroectodermal tumor of the central nervous system (CNS-PNET)” according to the World Health Organization (WHO) diagnostic lexicon of 2007 [1]. The designation “CNS-PNET” has been removed from the updated fourth edition of the WHO classification of CNS tumors; however, this new molecular entity is not yet included [2]. It is unclear whether CNS HGNET-BCOR should be classified in the category of “mesenchymal, nonmeningothelial tumors” or in that of “embryonal tumors.”

HGNET-BCOR is a highly aggressive entity which can metastasize outside the brain [3,4,5,6]. HGNET-BCOR primarily affects children and is characterized by somatic internal tandem duplications (ITDs) within the C-terminus of the BCL-6 co-repressor (*BCOR*) gene and by *BCOR* overexpression [1]. The same duplication has also been described in clear cell sarcoma of the kidney [7], soft tissue undifferentiated round cell sarcoma of infancy (URCSI), and primitive myxoid mesenchymal tumor of infancy (PMMTI) [8]. Preliminary survival data suggest that the CNS HGNET-BCOR entity has poor overall survival [1], but no standard therapy protocols exist for this tumor. 

We have previously described a personalized therapy protocol including elements from the pediatric rhabdoid and soft-tissue sarcoma protocol, radiation, and Arsenic trioxide to treat a pediatric HGNET-BCOR patient, achieving a complete remission that lasted for six months [5]. However, the tumor cells became resistant to the regimen, highlighting the need to identify new treatment approaches for this tumor.

The insulin-like growth factor (IGF) 2 is overexpressed in HGNET-BCOR [1,5]. IGF2 acts via the IGF1 receptor (IGF1R) promoting cell proliferation. The IGF pathway regulates cellular growth, proliferation, and survival. It is important in the development of several pediatric cancers, including sarcoma, glioma, neuroblastoma, medulloblastoma (MB), and Wilms tumor [9]. Different strategies have been tested to overcome IGF1R signaling, including IGF1R blockade by monoclonal antibodies, small-molecule tyrosine kinase inhibitors of IGF1R, and ligand-neutralizing strategies [10]. In spite of promising preclinical data, IGF1R inhibitors have not had success as single agents in clinical trials, and no formally approved drugs are available. However, compounds developed to inhibit other kinases, like ceritinib, a highly potent inhibitor of anaplastic lymphoma kinase (ALK), have shown off-target activity on IGF1R and may become relevant for the development of therapy protocols targeting IGF1R [11].

Good preclinical HGNET-BCOR models are needed to evaluate standard and novel treatment options. The only available preclinical model to date is the human primary cell culture PhKh1 that we have generated in our laboratory from the extracranial inoculation metastasis of a pediatric HGNET-BCOR patient (P1) [5,12]. Here, we further characterize PhKh1 cells and use them to determine their sensitivity to chemotherapy agents and to IGF1R inhibition to support the development of novel treatment approaches.

## 2. Results

### 2.1. Characterization of the HGNET-BCOR in Vitro Model PhKh1

We have previously described a HGNET-BCOR primary culture (PhKh1) isolated from the extracranial metastasis of a HGNET-BCOR patient [5,12]. The PhKh1 cells share similar features to the metastatic tissue of origin, including the presence of the BCOR ITD, the overexpression of *BCOR*, and the activation of the SHH and WNT pathways [12]. Here, we further characterized the cells by western blot analysis, DNA methylation analysis, copy number profile, and electron microscopy (Figure 1). To assess if *BCOR* is also translated into a protein, we performed western blot analysis with nuclear and cytosolic fractions of PhKh1 cells. A band of about ~200 kDa corresponding to the expected size of BCOR was detected mainly in the nuclear fraction (Figure 1A), according to the expected function of BCOR as a transcriptional regulator [13]. We then performed DNA methylation analysis using a chip-based assay and the brain tumor classification tool, recently described by Capper et al. (classifier version v11b4) [14]. DNA methylation profiling is highly robust and reproducible, and such profiles have been widely used to classify CNS tumors. PhKh1 cells and the metastatic tissue of P1 that was used to generate PhKh1 cells were both classifiable as HGNET-BCOR by 850k DNA methylation analysis. Chromosomal aberrations were similar between PhKh1 cells and the metastatic tumor tissue used to isolate the cells, being characterized mainly by a gain of chromosome (chr) 7 and the loss of a part of chr 11 (Figure 1B,C). No recurrent copy number changes have been described so far in HGNET-BCOR, and most cases show a flat profile in copy number analysis [1,3]. These data, however, were derived from the analysis of primary tumors, and gain of chr 7 and loss of a part of chr 11 may be features of more aggressive and metastasizing HGNET-BCOR tumor cells. Finally, electron microscopy of PhKh1 cell revealed that they have a size of about 8 µm and have the ability to phagocytize other cells (Figure 1D).

Taken together, our data demonstrate that PhKh1 cells have an aggressive phenotype, recapitulate features of the tumor of origin, and can be therefore used as an in vitro model to study HGNET-BCOR biology and to perform preclinical tests.

### 2.2. IGF2 and IGF1R are Highly Expressed in HGNET-BCOR

The expression of *IGF2* and *IGF1R* in HGNET-BCOR has been suggested previously by experiments based on Affymetrix arrays and RNA seq analysis [1,5]. Here, we validated the expression of *IGF2* and *IGF1R* by qRT-PCR in the tumors isolated from three HGNET-BCOR patients. The three patients had BCOR ITDs of different lengths (Figure 2A). As a control, material derived from a hepatoblastoma tumor was used (P4). Hepatoblastoma have a high *IGF2* expression due to genetic and epigenetic alterations in the *IGF2-H19* region [15]. Normal liver was used as a reference. Two HGNET-BCOR samples (P1 and P2) showed expression of *IGF2* comparable to that in the hepatoblastoma sample (Figure 2B). In one HGNET-BCOR sample, the expression was even stronger (Figure 2B, P3). The expression of *IGF1R* in all HGNET-BCOR tumors was stronger than in the hepatoblastoma sample (Figure 2C). The expression of IGF1R was confirmed at the protein level for the P1 tumor (Appendix A). High expression of *IGF2* and *IGF1R* was maintained in PhKh1 cells (Figure 2D).

These results confirm the assumption that the upregulation of *IGF2* and *IGF1R* is a common feature of HGNET-BCOR.

### 2.3. HGNET-BCOR is Responsive to Actinomycin D, Vinca Alkaloids, and IGF1R Inhibitors

For our drug screening in the PhKh1 model, we selected 11 compounds, including chemotherapeutics that we previously selected for our personalized treatment of HGNET-BCOR and three IGF1R kinase inhibitors, namely, linsitinib [16], picropodophyllin [17], and PQ401 [18]. All compounds were tested at concentrations in the range of 1–100,000 nM and were ranked by their respective IC_50_ values (Table 1 and Appendix A). Among the top hits in our drug screen were approved compounds in pediatrics, such as actinomycin D, vincristine, vinblastine, and doxorubicin. Platinum derivatives and etoposide were only effective at very high concentrations. PhKh1 cells were sensitive to inhibition with the kinase inhibitors linsitinib (Figure 3A) and picropodophyllin (Figure 3B), with an IC_50_ < 1µM, but not to PQ401 (Figure 3C). Because kinase inhibitors may be non-selective, we further analyzed the dependency of PhKh1 proliferation on IGF1R by using an IGF1R blocking antibody. Incubation of PhKh1 cells with the Alpha IR-3 neutralizing monoclonal antibody [19] significantly reduced the proliferation of PhKh1 cells in a concentration-dependent manner (Figure 3D). The proliferation of DAOY cells, an MB cell line used as a negative control, was not affected by the treatment with the blocking antibody against IGF1R, in accordance with a previous report [20].

Altogether, our drug screen showed that some of the conventional chemotherapeutic agents used in pediatric neuro-oncology are cytotoxic to PhKh1 cells at low concentrations and that IGF1R is a new potential target for therapy.

### 2.4. HGNET-BCOR is Responsive to Ceritinib

Because no specific IGF1R inhibitors are approved by any country’s medicines regulatory agency, we searched for small molecules with an off-target activity on IGF1R. Ceritinib is approved for ALK-positive lung cancer and can also inhibit IGF1R [11]. We first tested if the kinase domain of ALK was mutated in the PhKh1 cells, but we could not detect any mutation (data not shown). Ceritinib was able to inhibit the proliferation of PhKh1 cells with an IC_50_ of 0.31 µM (Table 1 and Figure 4A). Whereas many traditional chemotherapeutic agents inhibit proliferation in a timeframe of hours or a few days of treatment, targeted therapies that affect cancer-relevant pathways can require a longer time period to impact cellular growth and survival. To study the long-term effect of IGF1R inhibition, we incubated PhKh1 cells for 9 days with linsitinib or ceritinib. At the IC_50_ concentration, both drugs significantly inhibited the growth of the cells (Figure 4B). Notably, linsitinib and ceritinib at a concentration of 1 µM were able to stop the proliferation of PhKh1 cells. The 1 µM concentration was selected because it reflects a concentration that can be achieved in the plasma of patients [21,22]. 

In summary, ceritinib at a clinically achievable concentration is a potent and specific growth inhibitor of HGNET-BCOR cells.

### 2.5. Ceritinib Acts Via the IGF1R/AKT Pathway

For signaling transmission, IGF1R has to be phosphorylated. A cluster of three tyrosine residues, located at position Tyr1131, Tyr1135, and Tyr1136 within the kinase domain, is critical for receptor autophosphorylation [23]. In PhKh1 cells grown under serum starvation, IGF1R was not phosphorylated (Figure 5 lane 1). IGF2 stimulation induced the phosphorylation of IGF1R (Figure 5 lane 2). Ceritinib was able to inhibit the phosphorylation of IGF1R (Figure 5, lane 3). DMSO, which was used to dissolve ceritinib, had no effect on IGF1R phosphorylation (Figure 5, lane 4). As expected, linsitinib was also able to inhibit IGF1R phosphorylation (Appendix A).

IGF2 signals through the IGF1R to activate the PI3K/AKT/mTOR or the RAS/RAF/MEK/ERK pathway [24]. Ceritinib was able to block the phosphorylation of AKT (Figure 5, lane 3). No effect of ceritinib on ERK phosphorylation was observed (Figure 5, lane 3). 

These data indicate that in HGNET-BCOR, ceritinib can affect cell proliferation by inhibiting the phosphorylation of IGF1R and its downstream effector AKT.

### 2.6. Figures

**Figure 1 ijms-20-03027-f001:**
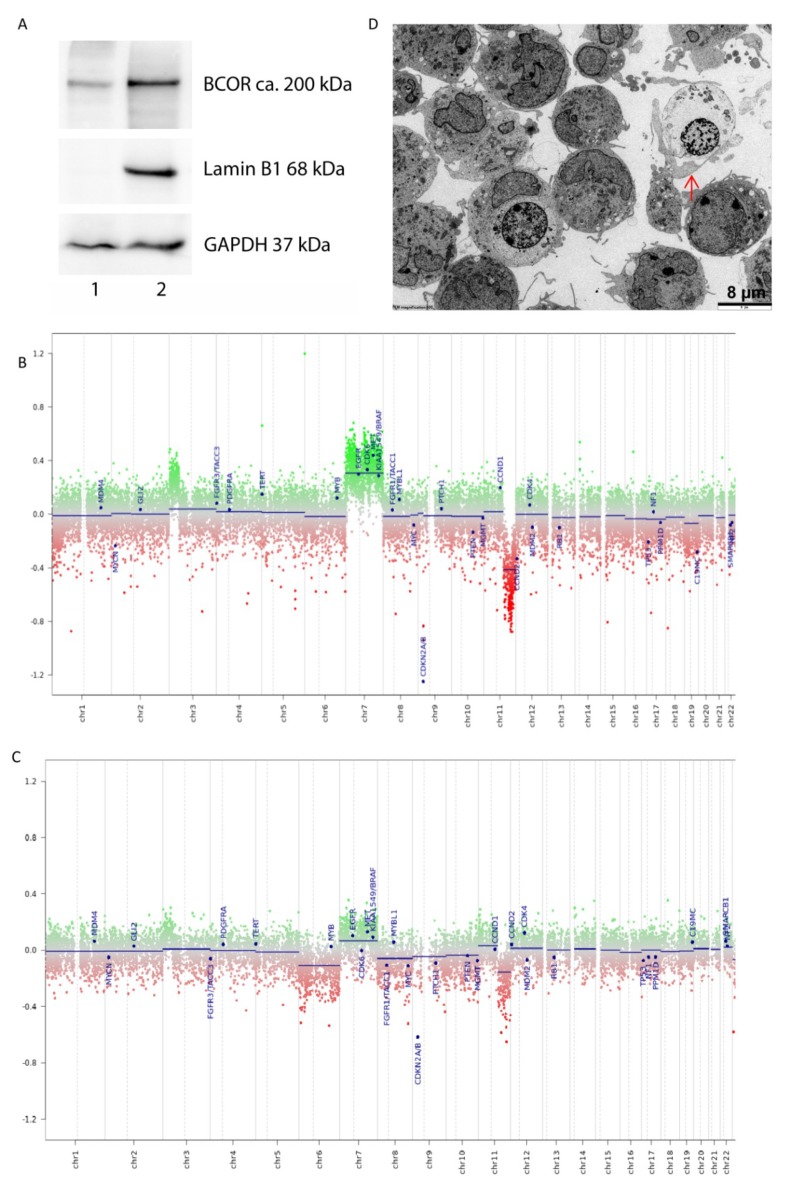
Characterization of a HGNET-BCOR *in vitro* model: (**A**) Cytosolic (lane 1) and nuclear (lane 2) subcellular components were extracted from PhKh1 cells and analyzed by western blot with a BCOR specific antibody or antibodies against different cellular compartments (Lamin B1 for nuclei, GAPDH for cytoplasm). DNA copy-number alterations of the PhKh1 cells (**B**) and of a metastasis of the HGNET-BCOR patient P1 (**C**,**D**). Electron microscopy of the PhKh1 cells at low passage. Cells were able to form branched projections and potentially have the ability to phagocyte other cells (red arrow). EM magnification 800.

**Figure 2 ijms-20-03027-f002:**
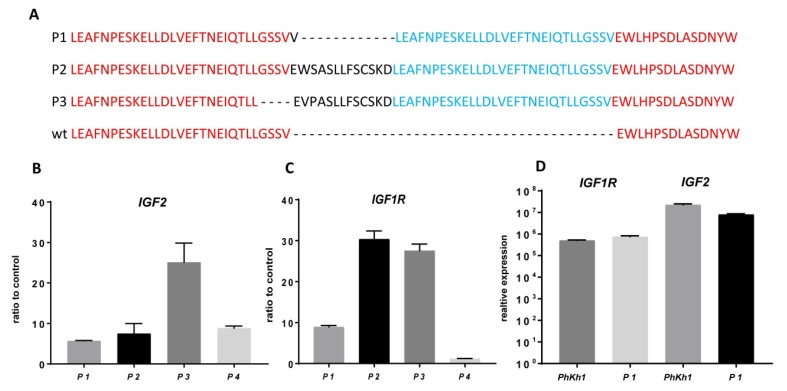
*IGF2* and *IGF1R* are highly expressed in high-grade neuroepithelial tumor with BCOR alteration (HGNET-BCOR). (**A**) Protein sequence of the *BCOR* allele of patients P1, P2, and P3 carrying BCOR internal tandem duplications (ITD) compared to wild-type (wt) *BCOR*. (**B**,**C**) qRT-PCR analysis was performed using primers specific for *IGF2* and *IGF1R*. Data are shown as the fold change of the expression in the tumor tissues of three HGNET-BCOR patients (P1, P2, and P3) and one hepatoblastoma patient (P4) with respect to the expression in normal liver (control). Expression analysis was done in triplicate on RNA extracted from formalin-fixed paraffin-embedded (FFPE) material. (**D**) qRT-PCR analysis was performed using primers specific for *IGF2* and *IGF1R*, with RNA extracted from PhKh1 cells or frozen material from the metastasis of origin of PhKh1 cells. After normalization to the expression of the housekeeping gene *HPRT1*, the relative quantification value was expressed as 2−ΔΔCt. The calibrator was calculated as the maximal number of cycles used in the PCR (40) minus the mean of the *HPRT1* Ct values, resulting in a value of 19.

**Figure 3 ijms-20-03027-f003:**
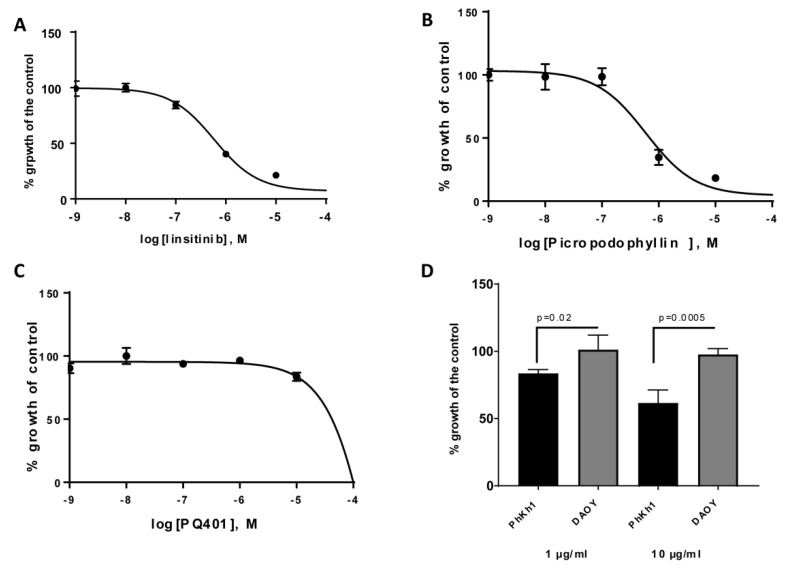
HGNET-BCOR is sensitive to IGF1R inhibition: (**A**–**C**) PhKh1 cells were treated with the indicated IGF1R inhibitors at concentrations from 1 nM to 100 μM. The logarithm of the molarity (M) is displayed on the X-axis. The percent of viable cells compared to the control treated with vehicle alone is shown on the Y-axis. The data were fitted to a sigmoidal dose–response curve using GraphPad software. A representative experiment of three independent experiments is shown. (**D**) PhKh1 and DAOY cells were incubated with a blocking anti-IGF1R antibody at the indicated concentration, and cell proliferation was measured. The percent of viable cells compared to the control treated with vehicle alone is shown on the Y-axis. Statistics were performed using Student’s *t*-test. A representative experiment of three independent experiments is shown.

**Figure 4 ijms-20-03027-f004:**
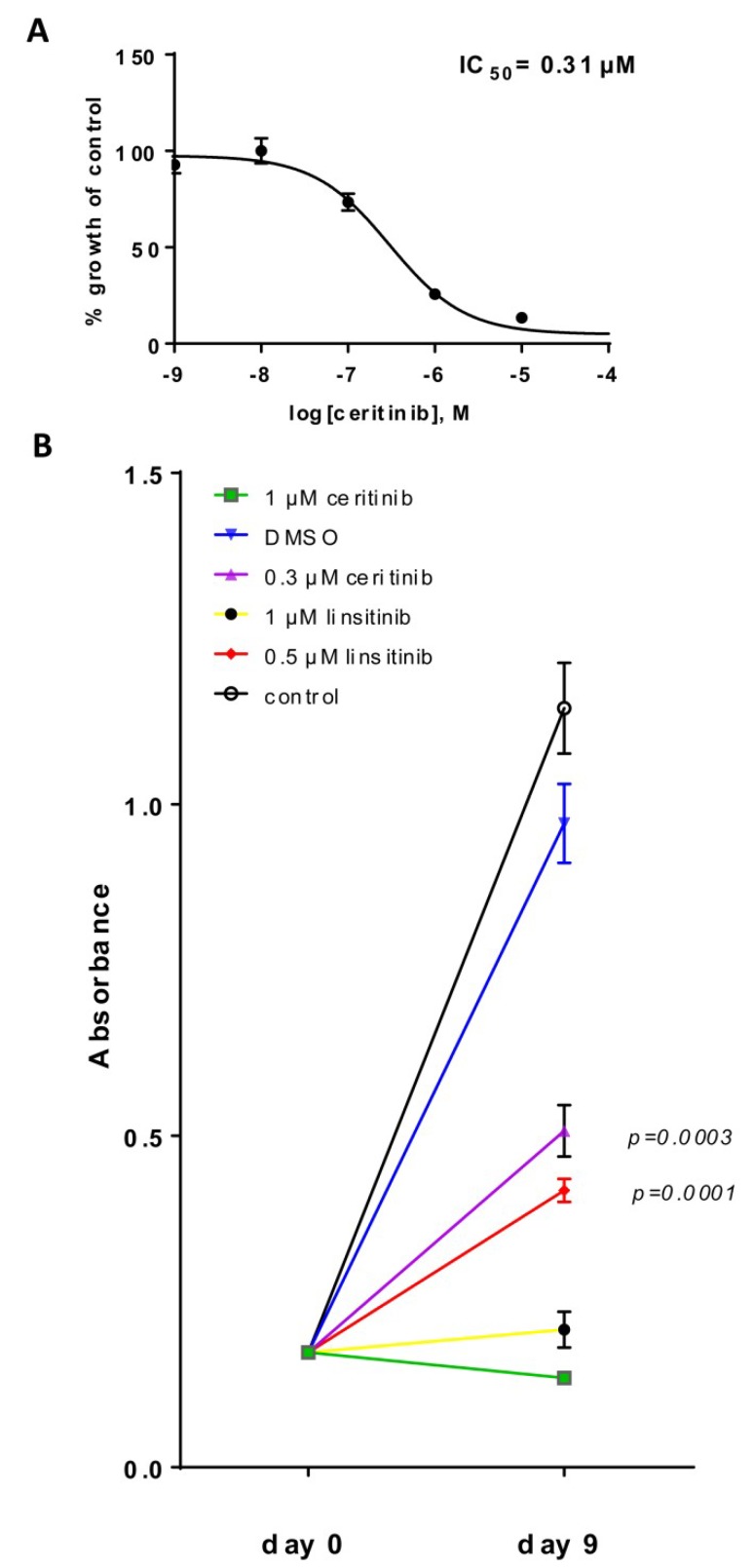
HGNET-BCOR is sensitive to the inhibition with ceritinib: (**A**) PhKh1 cells were treated with ceritinib at concentrations from 1 nM to 100 μM. The X-axis shows the logarithm of M, and the Y-axis the percent of viable cells compared to the control cells treated with DMSO. The GraphPad software was used to create the dose–response curve. A representative experiment of three independent experiments is shown. (**B**) PhKh1 cells were grown for 9 days in the presence of ceritinib, linsitinib, vehicle alone (DMSO), or without any treatment (control) at the indicated concentrations. Absorbance after incubation with the WST-1 reagent is indicated. Statistics were performed using Student’s *t*-test at day 9 compared to the DMSO control. A representative experiment of three independent experiments is shown.

**Figure 5 ijms-20-03027-f005:**
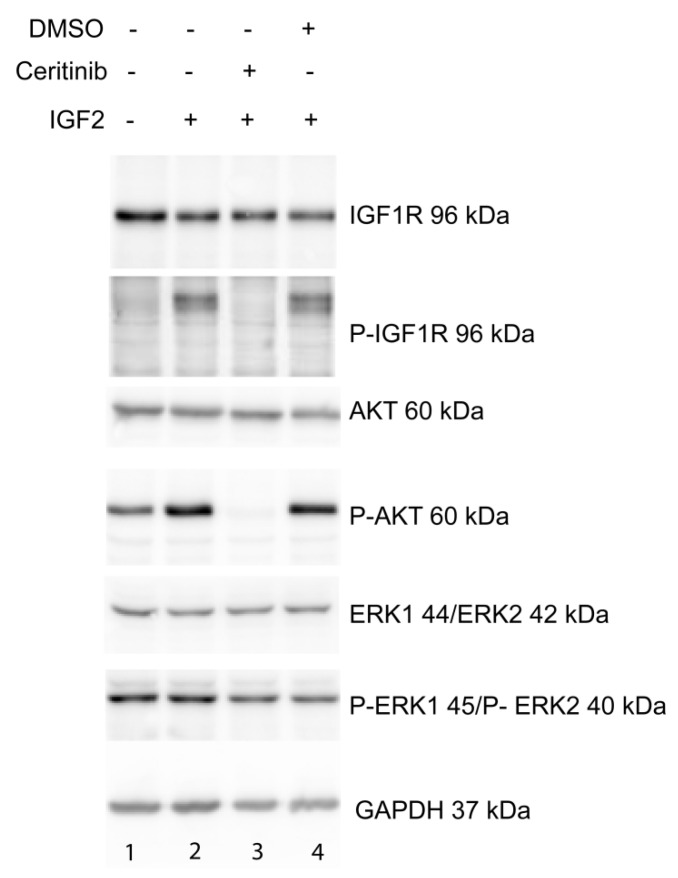
Ceritinib acts via the IGF1R/AKT pathway: PhKh1 cells were stimulated after starvation with IGF2 in the presence or absence of ceritinib or vehicle (DMSO). The expression of IGF1R, AKT, ERK1/ERK2, and their respective phosphorylated forms (P-IGF1R, P-AKT, P-ERK1/P-ERK2) was analyzed by western blot with specific antibodies. GAPDH was used as loading control. A representative experiment of three independent experiments is shown.

## 3. Discussion

HGNET-BCOR is a highly malignant tumor with a poor prognosis [1,3,5] that is in need of new therapeutic approaches. HGNET-BCOR has an aggressive phenotype and can develop extra-CNS metastases [3,4,6,12]. The HGNET-BCOR model we used here to identify new therapeutic drugs consists of primary cells isolated from extracranial metastases of a HGNET-BCOR patient and represents a particularly aggressive form of the disease. The malignant nature of the metastatic HGNET-BCOR cells is underlined by the ability of PhKh1 cells to phagocytose other cells. This feature has been described for highly aggressive cells with high metastatic potential [25]. Cannibalism is a key survival option for malignant cancers [26]. Because of the dependence of the phagocytosis process on the microtubule cytoskeleton, it might be speculated that microtubule inhibitors like vincristine and vinblastine can act as effective drugs. Our data show indeed the potential of *Vinca* alkaloids for the development of new therapeutic protocols. However, while vincristine is associated with highly variable neurotoxicity that often necessitates dose reductions, thereby compromising efficacy, vinblastine monotherapy has shown promising activity and a low-toxicity profile in patients with pediatric low-grade glioma [27]. Doxorubicin and actinomycin D were also effective in inhibiting PhKh1 cell proliferation but they are generally not used in the treatment of brain tumors or metastasis in the brain because of their low penetration across the blood–brain barrier. However, they could be relevant for the treatment of HGNET-BCOR patients with extracranial metastases. 

Activation of the IGF2 pathway has been described in several pediatric tumor entities, and preclinical findings using inhibitors of the IGF axis have demonstrated antitumor activity [28]. In this work, we show that HGNET-BCOR tumors expressed high levels of *IGF2* and that the proliferation of the primary HGNET-BCOR cells PhKh1 could be reduced by targeting the IGF1R receptor on the plasma membrane with a blocking antibody or with intracellular kinases inhibitors. To date, neither monoclonal antibodies nor small-molecule tyrosine kinase inhibitors directed against IGF1R have been approved, but drug repurposing may represent a way forward to rapid clinical use. Ceritinib gained US Food and Drug Administration approval in 2014 for the treatment of patients with ALK-positive metastatic non-small cell lung cancer (NSCLC) who have progressed on or are intolerant to crizotinib. Ceritinib inhibits the kinases of proto-oncogene receptor tyrosine kinase (ROS1) and IGF1R, although it is most active against ALK [11]. In our model of HGNET-BCOR, ceritinib was able to affect cell proliferation and to inhibit the phosphorylation of IGF1R. Notably, the concentration of 1 µM ceritinib that we used in long-term experiments can be achieved in the plasma of patients [22]. Lung cancer patients with brain metastases also respond to the treatment with ceritinib, suggesting that this drug is effective across the blood–brain barrier [29]. The concentration achieved in the mouse brain after oral administration was low, with a brain-to-plasma ratio below 0.3 [30], but the blood-to-brain exposure ratio of ceritinib in humans has yet to be determined. The inhibition of P-glycoprotein (P-GP/ABCB1) and breast cancer resistance protein (BCRP/ABCG2) may improve the accumulation of ceritinib in the brain [30]. A phase I study (NCT01742286) in pediatric patients with ALK-aberrant malignancies has shown that ceritinib can be given to pediatric patients, with acceptable toxicities, especially if the drug is taken with food.

IGF1R can act via the AKT and ERK1/ERK2 pathways to promote cell growth, differentiation, and proliferation. Ceritinib was able to reduce the phosphorylation of AKT in HGNET-BCOR, suggesting that a combination of ceritinib with mTOR inhibitors may potentiate the treatment. The inhibition of the phosphorylation of IGF1R and AKT after incubation with ceritinib has been described in rhabdomyosarcoma, and a synergistic effect of ceritinib and dasatinib was observed [31]. Whether this synergy also exists in HGNET-BCOR remains to be elucidated.

We have previously shown that targeting the SHH pathway using arsenic trioxide (ATO) in combination with irradiation can induce a complete remission in a pediatric patient, but resistance to this therapy developed [5]. IGF2 is an SHH pathway-responsive molecule in some cell types [19,32], and the activation of the SHH pathway in HGNET-BCOR may explain the high expression of *IGF2*. However, the regulation of *IGF2* by the SHH pathway is complex, and other not yet identified regulators may cooperate with the SHH pathway to induce *IGF2* expression in HGNET-BCOR. Other mechanisms such as changes in the 3’ untranslated region of *IGF2* [33] and altered methylation of the *IGF2* promoter should be also considered [15]. Whether the SHH and IGF pathways act synergistically to support HGNET-BCOR proliferation remains to be evaluated. In the case of a synergy, co-targeting of both pathways through a combination of ATO and ceritinib could improve the efficacy of a targeted therapy.

In conclusion, we have demonstrated that HGNET-BCOR is very sensitive to inhibition by *Vinca* alkaloids, doxorubicin, actinomycin D, and ceritinib. All these drugs are already approved or in clinical trials for malignancies in pediatric oncology, and priority should thus be given to validate them first in animal models and finally in multimodal therapies in future clinical trials for HGNET-BCOR patients. Finally, the use of the off-target IGF1R inhibitor ceritinib may accelerate the development of clinical protocols for the treatment of tumor entities driven by IGF2 and IGF1R. 

## 4. Materials and Methods 

### 4.1. Patient (P) Tissue Samples

P1, P2, and P3 are HGNET-BCOR patients; P1 and P3 are male, and P2 is a female patient. The clinical history of P1 and P2 has already been described [11]. P3 is a male patient diagnosed with HGNET-BCOR in the left temporal lobe at the age of two years. P4 is a five-year-old male child with hepatoblastoma. Normal liver tissue was obtained from a five-year-old male child with no liver pathology, used as a control. This study was performed in agreement with the declaration of Helsinki. In accordance with the ethics committee of Rhineland-Palatinate, the patients’ parents agreed with the scientific use of the surplus material. The patient samples were collected and used in the study after obtaining informed consents from the patients’ parents.

### 4.2. Cells

The primary PhKh1 cells of HGNET-BCOR were previously described [5,11]. Only low-passage cells were used in the experiments. DAOY cells were obtained directly from a cell bank that performs cell line characterizations (ATCC) and passaged for fewer than 6 months after receipt. PhKh1 cells were maintained in a humidified incubator with 5% CO2 at 37 °C and cultured to a confluency of 70–80% in advanced DMEM (GibcoTM, Thermo Fisher Scientific, MA, USA) supplemented with 10% Human Serum (Sigma-Aldrich Co., MO, USA), 2 mM L-Glutamin (Sigma-Aldrich Co., MO, USA), and Penicillin–Streptomycin (GibcoTM, Thermo Fisher Scientific, MA, USA) to final concentrations of 100 U/mL and 100 µg/mL, respectively. DAOY cells were cultivated under the same conditions as PhKh1 cells, with the difference that the serum component was 10% fetal calf serum (Gibco™, Thermo Fisher Scientific, MA, USA) instead of 10% human serum.

### 4.3. Nucleic Acid Extraction 

DNA was extracted using the QIAamp DNA FFPE kit (Qiagen, Hilden, Germany). RNA extraction was performed using the RNeasy FFPE Kit for FFPE material (Qiagen). RNA was converted to cDNA using the PrimeScript RT Reagent Kit with gDNA Eraser (Takara Bio Europe, Saint-Germain-en-Laye, France). Quality control was performed using a 2100 Bioanalyzer (Agilent Technologies, Waldbronn, Germany). Because of the expected low quality of the RNA extracted from FFPE material, the protocol for cDNA synthesis was changed. Instead of utilizing the RT Primer Mix that is included in the kit, we used gene-specific reverse primers, and the reaction samples were incubated for 60 min at 42 °C instead of 15 min at 37 °C. 

### 4.4. RT-PCR and qRT-PCR

qRT-PCR was performed using the LightCycler 480 II Detection System and Software (Applied Biosystems, Darmstadt, Germany) with KAPA SYBR FAST LightCycler 480 Kit (PeqLab, Erlangen, Germany). The following primers were used: *IGF2*: 5’-GACCGTGCTTCCGGACA and 5’-TCGAGCTCCTTGGCGAGC; *IGF1R*: 5’-GGCCTTCTGGACAAGCCAG and 5’-AGACCTCCCGGAAGCCAG; *HPRT*: 5’-TGACACTGGCAAAACAATGCA and 5’-GGTCCTTTTCACCAGCAAGCT. 

### 4.5. DNA Sequencing

The coding region of ALK containing the kinase domain was analyzed from the cDNA using primers described in [34]. The region containing the BCOR ITD was amplified using the primers 5’-GGAAATTGTCACCATTGCAGAGG and 5’-TGTACATGGTGGGTCCAGCT. Sanger Sequencing was performed as previously described [11].

### 4.6. In Vitro Drug Screening

All inhibitors were commercially purchased (Selleck Chemicals, TX, USA) and dissolved in DMSO (Sigma-Aldrich Co., MO, USA) at a concentration of 10 mM, with the exception of actinomycin D which was dissolved at a concentration of 1 mM. The cells were plated in triplicate at a density of 5000 cells/well in a 96-well plate; each plate contained a nontreatment condition, vehicle condition, and blank. Viable cells were quantified using the WST-1 reagent (Roche, Mannheim, Germany). Dose–response curves were plotted after 72 h to determine the half-maximal inhibitory concentration (IC_50_) using the GraphPad Prism v.5 (GraphPad Software, San Diego, CA, USA). For IGF1R blockade, the cells were incubated with an anti-IGF1R blocking antibody (clone αIR3, Merck Millipore, Darmstadt, Germany) for 48 h at a concentration of 1 µg/mL or 10 µg/mL. For long-term experiments, 5000 cells were plated in triplicate and incubated with different concentrations of ceritinib, linsitinib, or vehicle alone for 9 days. Statistics were performed using Student’s *t-*test.

### 4.7. Phosphorylation Assay 

PhKh1 cells were cultured in DMEM (high glucose, no glutamine, no phenol red) (GibcoTM, Thermo Fisher Scientific, MA, USA) which was stripped with dextran-coated charcoal (Sigma-Aldrich Co., MO, USA). Dextran-coated charcoal-stripped DMEM was supplemented with 10% human serum (Sigma-Aldrich Co., MO, USA), 2 mM L-Glutamin (Sigma-Aldrich Co., MO, USA), 1 mM Sodium Pyruvate (GibcoTM, Thermo Fisher Scientific, Massachusetts, USA), and Penicillin–Streptomycin (GibcoTM, Thermo Fisher Scientific, MA, USA) to final concentrations of 100 U/mL and 100 µg/mL, respectively. IGF2 (R&D Systems Inc., MN, USA) was dissolved in sterile PBS (Sigma-Aldrich Co., MO, USA) to a final concentration of 100 µg/mL. Three million PhKh1 cells were plated into 10 cm culture dishes and grown in charcoal-stripped DMEM for 1 day. The monolayers were washed with PBS and serum-starved overnight. The cells were treated with 1 µM ceritinib, 1 µM linsitinib, or vehicle alone for 2 h, followed by incubation with 20 ng/mL IGF2 for 15 min. After washing the cells with PBS, lysis ensued with 50 mM Tris·HCl, pH 8.0, 150 mM sodium chloride, 5 mM magnesium chloride, 1% Triton X-100, 0.5% sodium deoxycholate, 0.1% SDS, 40 mM sodium fluoride, 1 mM sodium orthovanadate, and the cOmplete protease inhibitor cocktail (Roche Diagnostics, Indianapolis, IN, USA). Lysates were analyzed by Western blot

### 4.8. Preparation of Subcellular Compartments and Lysis of Tumor Tissue 

PhKh1 cells were cultured as described above. Nuclear and cytosolic extracts were generated according to Holden and Horton [35]. The cells were lysed in buffer 1 (150 mM NaCl, 50 mM HEPES pH 7.4, 1% NP-40, cOmplete protease inhibitor cocktail for 1 h at 4 °C on a shaker. After centrifugation at 7000 rcf for 15 min at 4 °C, the supernatant containing the cytoplasm was isolated, and the pellet containing the nuclei was resuspended in buffer 2 (150 mM NaCl, 50 mM HEPES pH 7.4, 0.5% sodium deoxycholate, 0.1% sodium dodecyl sulfate, cOmplete protease inhibitor cocktail) and incubated at 4 °C for 1 h. After centrifugation and ultrasound treatment, the supernatant contained proteins of the nuclear fraction. Tumor tissue was disrupted using the TissueLyser II (Qiagen, Hilden, Germany) and lysed with 50 mM Tris·HCl, pH 8.0, 150 mM sodium chloride, 5 mM magnesium chloride, 1% Triton X-100, 0.5% sodium deoxycholate, 0.1% SDS, 40 mM sodium fluoride, 1 mM sodium orthovanadate, and cOmplete protease inhibitor cocktail) for 1 h at 4 °C. After centrifugation, the supernatant contained the extracted proteins. Protein concentration of the samples was measured by Bradford Assay.

### 4.9. Western Blot Analysis

The lysates were loaded on SDS-PAGE gels, followed by blotting onto polyvinylidene difluoride membranes (BioRad Laboratories, Inc., CA, USA). The following antibodies were obtained from Cell Signaling Technology (MA, USA): GAPDH (14C10) (cat # 2118, 1:1000 dilution), Lamin B1 (D4Q4Z) (cat# 12586, 1:1000 dilution) (Cell Signaling Technology), IGF1 Receptor β (D23H3) XP® (cat # 9750, 1:1000 dilution), Phospho-IGF1 Receptor β (Tyr1131)/Insulin Receptor β (Tyr1146) (cat # 3021, 1:1000 dilution), AKT (cat # 9272, 1:1000 dilution), Phospho-AKT (Ser473) (cat # 9271, 1:1000 dilution). The antibodies ERK1/ERK2 (cat # MAB1576, 0.5 µg/mL dilution) and Phospho-ERK1 (T202/Y204)/ERK2 (T185/Y187) (cat # MAB1018, 0.5 µg/mL dilution) were obtained from R&D Systems Inc. (MN, USA). The anti-BCOR antibody (cat# ab88112, 0.5 µg/mL dilution) was purchased from Abcam (Cambridge, UK). The HRP-linked secondary antibodies were obtained from Cell Signaling Technology (MA, USA) (anti-mouse IgG (cat# 7076)) and Sera Care (MA, USA) (anti-rabbit IgG (cat# 074-1516)). Detection was done by SuperSignal™ West Dura Extended Duration Substrate (Thermo Fisher Scientific, MA, USA), and imaging was performed on Fusion Pulse TS (Vilber Lourmat, Eberhardzell, Germany). 

### 4.10. Transmission Electron Microscopy (TEM)

Transmission electron microscopical analysis was performed with an EM 410 (Phillips). Samples of 0.5 × 0.5 cm were fixed in buffered glutaraldehyde (2.5%) overnight, post-fixed with OsO4, and then embedded in Agar 100 (Plano), which was let polymerize for at least 24 h. Ultrathin sections were cut with the Ultracut E microtome (Leica). Visualization took place at 80 KV, and images were photodocumented.

### 4.11. DNA Methylation Analysis

DNA methylation analyses were performed using an EPIC 850k DNA methylome chip (Illumina, San Diego, USA). We used standard protocols for tissue and DNA processing. Hybridization and processing of the chips were performed as indicated by the manufacturer. Data were preprocessed using Illumina Genome Studio, and further analysis was performed after uploading raw data (idat files) onto the platform MolecularNeuroPathology.org [14].

## Figures and Tables

**Table 1 ijms-20-03027-t001:** Overview of chemotherapeutics and insulin-like growth factor receptor (IGF1R) inhibitors tested in vitro on PhKh1 cells. The IC50, class of the drug, its mechanism of action, and tumor entity generally treated with the drugs are indicated. FDA: Food and Drug Administration.

Drug	IC_50_ nM (±SD)	Class	Mechanism of Action	Entity
Actinomycin D (*n* = 3)	<1	Antibiotic	Intercalation into DNA	Sarcoma
Vinblastine (*n* = 2)	<1	*Vinca* alkaloid	Binds tubulin	Sarcoma
Vincristine (*n* = 3)	6.4 (±4)	*Vinca* alkaloid	Binds tubulin	Sarcoma, Brain tumors
Doxorubicin (*n* = 4)	89.3 (±65)	anthracycline	Intercalation into DNA	Sarcoma
Ceritinib (*n* = 5)	277 (±99)	Kinase inhibitor	ALK, ROS1, IGF1R inhibitor	Lung Cancer
Linsitinib (*n* = 3)	516.7 (±89)	Kinase inhibitor	IGF1R Inhibitor	Not yet FDA-approved
Picropodophyllin (*n* = 3)	475.1 (±111)	Kinase inhibitor	IGF1R Inhibitor	Not yet FDA-approved
Etoposide (*n* = 3)	5808 (±795)	Derivative of podophyllotoxin	Forms complex with topoisomerase II	Sarcoma
Cisplatin (*n* = 3)	6976 (± 3100)	Platin derivative	Binds to DNA	Sarcoma, Brain tumors
Carboplatin (*n* = 3)	>10,000	Platin derivative	Binds to DNA	Brain tumors
PQ401 (*n* = 3)	>10,000	Kinase inhibitor	IGF1R Inhibitor	Not yet FDA-approved

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
