# Peer review of "IGF1R Is a Potential New Therapeutic Target for HGNET-BCOR Brain Tumor Patients"

_ijms, 2019, doi:10.3390/ijms20123027_

Round 1

Reviewer 1 Report

The manuscript is an interesting study focused on the involvement of IGF1R in the pathogenesis of HGNET-BCOR brain tumor. Even if the number of tumor sample is very small I consider that this study offers important information for the medical domain, and for the development of future therapies for this malignancy. Also, minor errors regarding the English language should be corrected. I recommend future in vivo experiments to validate these results.

Author Response

The reviewer suggested improving the spelling and commented on the need of in vivo models in the future to validate the in vitro data of this study. We have carefully proof-read spell and corrected some minor spelling errors (line 102, 247 and 359). Moreover we commented the validation in animal models in the discussion (line 285)

Reviewer 2 Report

The present work described the IGF2/IGF1R signaling in HGNET-BCOR brain tumor and claimed that IGF1R is a potential new therapeutic target for this are type of brain cancer. The manuscript was also overall well written, but a few comments for the authors’ consideration.

1.     Figure 2. The protein levels of IGF2 and IGF1R (phosphorylation) should be measured by western blot to validate the qPCR results.

2.     There is always a concern about the selectivity of pharmacological compounds used. The authors should consider using gene knockdown or knockout to validate the results from pharmacological inhibition.

3.     Figure 5. Selective IGF1R inhibitor like linsitinib should be included as positive control for comparison with ceritinib, which primarily acts on ALK.

Author Response

The reviewer suggested improving the results and conclusions by adding some more data.  We addressed the specific comments as follow:

1.     Figure 2. The protein levels of IGF2 and IGF1R (phosphorylation) should be measured by western blot to validate the qPCR results.

Unfortunately the material of P2 and P3 is available only as FFPE. For P1, we had enough fresh frozen material to analyze the expression of IGF1R and P-IGF1R at the protein level (Line 117, Line 369ff and Supplemental Figure 2). While a distinct expression of IGF1R is clearly detectable, the IGF1R phosphorylation is only barely detectable. This can be explained by the fact that tissue handling during surgery influences the phosphoprotein levels (PMID: 18667411) and no Kinase and phosphatase inhibitors were used to stabilize the activation status of proteins in our specimens.

2.     There is always a concern about the selectivity of pharmacological compounds used. The authors should consider using gene knockdown or knockout to validate the results from pharmacological inhibition.

We agree that pharmacological compounds and particularly TKI are not always selective. For this reason we have shown dependency of PhKh1 proliferation on IGF1R also using a blocking monoclonal antibody. We have discussed this point in the revision at line 130 ff.

3.     Figure 5. Selective IGF1R inhibitor like linsitinib should be included as positive control for comparison with ceritinib, which primarily acts on ALK.

We have added linsitinib as positive control (line 168ff and Supplemental Figure 2)

Round 2

Reviewer 2 Report

The manuscript has been revised adequately.